# Let Me Finish My Sentence:
# Video Temporal Grounding with Holistic Text Understanding

## ABSTRACT

Video Temporal Grounding (VTG) aims to identify visual frames in a video clip that match text queries. Recent studies in VTG employ cross-attention to correlate visual frames and text queries as individual token sequences. However, these approaches overlook a crucial aspect of the problem: a *holistic understanding* of the query sentence. A model may capture correlations between individual word tokens and arbitrary visual frames while possibly missing out on the global meaning. To address this, we introduce two primary contributions: (1) a visual frame-level gate mechanism that incorporates holistic textual information, (2) cross-modal alignment loss to learn the fine-grained correlation between query and relevant frames. As a result, we regularize the effect of individual word tokens and suppress irrelevant visual frames. We demonstrate that our method outperforms state-of-the-art approaches in VTG benchmarks, indicating that holistic text understanding guides the model to focus on the semantically important parts within the video.

## CCS CONCEPTS

• **Computing methodologies** → **Video summarization**; **Scene understanding**; **Activity recognition and understanding**; • **Information systems** → **Video search**.

## KEYWORDS

Video Temporal Grounding, Video Moment Retrieval, Video Highlight Detection

## 1 INTRODUCTION

Recently, the boom of video and streaming platforms such as Disney+, YouTube, TikTok, Netflix, etc., has led to an abundance of online video content. Naturally, the growing pool of videos from various platforms sparked an interest in efficiently searching videos using text inputs. Video Temporal Grounding (VTG) is a prominent research area within video-text search, that focuses on grounding visual frames that correspond to custom queries. Within VTG, various tasks such as Moment Retrieval (MR) [6, 14, 24, 27, 50, 51] and Highlight Detection (HD) [20, 37, 47] has been proposed.

The goal of MR is to identify time intervals that are highly relevant to text queries, while HD assesses the significance of each video frame to select the most significant segments. HD can be

*ACM MM, 2024, Melbourne, Australia*
© 2024 Copyright held by the owner/author(s). Publication rights licensed to ACM.
ACM ISBN 978-x-xxxx-xxxx-x/YY/MM
https://doi.org/10.1145/nnnnnnn.nnnnnnn

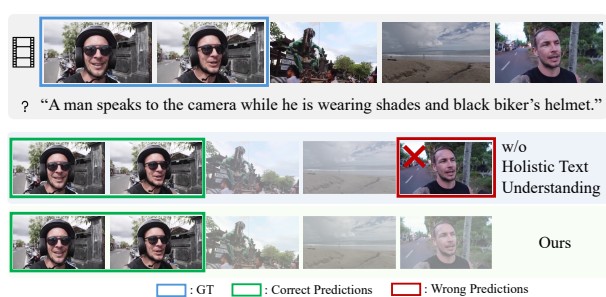

**Figure 1: This example shows the critical role of holistic text understanding in Video Temporal Grounding. Unlike previous works that do not take holistic text understanding into account, our method effectively filters out frames that do not correspond to the full context of the query. Here, our model does not predict the final frames due to the absence of the helmet and shades mentioned in the query.**

categorized into two perspectives: query-independent and query-dependent. This paper focuses exclusively on query-dependent HD, which utilizes text queries to analyze video content. Despite the distinct operational focuses of MR and HD, both tasks share the core aim of aligning video content with corresponding natural language queries. Recognizing their shared goal, [13] introduced the QVhighlights dataset. This allows for simultaneous training on both MR and HD, promoting a unified approach to VTG.

Prior approaches have focused on developing cross-modal interaction strategies [10, 18] or developed models specifically for the demands of the MR and HD tasks [36, 42]. However, despite these advancements, existing models often treat the text query as a sequence of tokens rather than an entire sentence. These approaches may neglect the overall textual semantics, as individual text tokens either lack the capacity to convey the complete meaning and/or cause the model to attend to unrelated words or frames. This oversight can limit the model's ability to fully capture the intent of the query. For instance, as shown in Fig. 1, the model without holistic text understanding may highlight the latter part of a video in response to the tokens inside the clause "A man speaks to the camera." To tackle this issue, we focus on utilizing the global information contained within text queries, emphasizing its importance in accurately identifying the most relevant video frames.

To this end, we propose a novel framework that utilizes a holistic, or *global* text anchor—representing the full input sentence—to selectively suppress less relevant video frames while emphasizing the relevant ones. Leveraging this specialized token, our framework introduces a gated cross-attention mechanism that effectively filters out irrelevant video content. The gated cross-attention employs two gating mechanisms: *local* and *non-local* gates. The local gate assigns weights based on channel-wise similarity between the text anchor and individual frames (frame-level). In contrast, the

non-local gate assigns weights by evaluating the overall relevance between the text anchor and the entire video (clip-level), prioritizing frames that exhibit greater contextual alignment with the global text query. The overall relevance is computed through an anchor-query cross-attention mechanism that employs the text anchor as a query to interact with video frames. An attention map derived from this interaction effectively assesses clip-level correlation, thereby emphasizing the focus on contextually pertinent frames. To further refine the precision in assessing similarity, we introduce two fine-grained alignment losses that optimize clip-level consistency and frame-level relevance. These losses use the text query as an anchor, enabling a more targeted and accurate alignment between the video content and the corresponding textual information. The clip-level consistency loss aims to minimize the discrepancy between the anchor and the clip-level video feature outputted from the anchor-query cross-attention layer. This reduction aims to enhance the accuracy in determining the relative significance of each video frame in relation to the anchor. On the other hand, the frame-level relevance loss aims to ensure that frames relevant to the text query are closely aligned with the global text query representation. It minimizes the distance between the global query and the corresponding frames, thus enhancing the model's ability to more accurately align relevant video content with the text.

To demonstrate the effectiveness of our proposed framework, we conduct extensive experiments on the QVHighlights [13] dataset, as well as on other notable VTG benchmarks, including Charades-STA [6] and TACoS [27]. Our experimental results show that our proposed method, to the best of our knowledge, achieves state-of-the-art performance compared to previous methods. Additionally, we carry out a detailed ablation study to further assess and validate the advantages of our proposed method. The contributions of our approach are summarized as follows:

- We introduce a novel framework that utilizes a *global text-anchor* based approach to enhance video grounding, leveraging holistic textual information to accurately filter and prioritize relevant frames.
- To best exploit our global text-anchor, we introduce two cross-attention mechanisms where we integrate both gated and anchor-query cross-attention for deeper video-text interaction.
- We propose fine-grained alignment loss functions (clip-level and frame-level) anchored by the text query, designed to refine the accuracy in measuring similarities and aligning video content with the given text queries.

## 2 RELATED WORKS

**Moment Retrieval** (MR) is a task that identifies temporal moments that are highly relevant to a given natural language query. Within MR, previous works have been categorized in two sections: proposal-based and proposal-free. As in the name, proposal-based methods have generally followed a pipeline where a model generates candidate windows from the entirety of the video, then grants a rank based on the matched scores. This approach often relies on predefined temporal structures like sliding windows [1, 6, 18, 31, 52] or temporal anchors [3, 33, 44, 46, 49, 51] to formulate candidate moments. On the other hand, proposal-free methods solve the task as

a regression problem where they directly regress the start and end time frames through various multimodal methods. Within this line of works, previous methods have utilized methods such as multi-modal coattention [7, 23, 48, 50], dynamic filters [29], and additional features [4, 29] to varying degrees of success. Both proposal-based and proposal-free methods are effective but rely on hand-crafted processes such as proposal generation and non-maximum suppression.

**Highlight Detection** (HD) is a task that focuses on the evaluation of individual video clips' significance by assigning them clip-wise saliency scores and highlighting the segments with the highest scores. However, HD datasets [30, 34, 38] are usually domain-specific and operate independently of textual queries. The common datasets [20, 47] available for query-based highlight detection offer a limited number of annotated frames for training and evaluation. The scarcity of query-dependent HD datasets underscores the perception of HD as primarily a vision-only problem. Unlike earlier HD datasets that were query-agnostic, we explore and test on an HD task that offers a saliency score for query-relevant clips, facilitating models to perform query-dependent highlight detection.

**Video Temporal Grounding** seeks to combine the two aforementioned tasks. Despite their similar objectives, the absence of a unified dataset supporting both tasks has constrained simultaneous exploration and combination of these two fields. To this end, [13] released QVhighlights and proposed Moment-DETR as a simple baseline. Subsequently, UMT [21] explored the addition of audio cues to enrich query generation, and QD-DETR [22] enhanced query-dependent video representations through a specialized cross-attention module. MH-DETR [45] explores further cross-modal integration by merging visual and textual features through a pooling mechanism, while UniVTG [17] proposes a unified grounding model including video summarization. More recent efforts by TR-DETR [36] and UVCOM [42] incorporate distinct task characteristics into their frameworks. Even with these advancements, existing models often interpret text queries as a sequence of individual tokens, neglecting the holistic semantics of the entire query. To address this issue, we introduce a novel framework that emphasizes holistic textual understanding to accurately identify and emphasize relevant video segments.

**Cross-modal alignment** is an important concept in tasks that involve multiple modalities. This type of alignment ensures that representations from various modalities, such as visual and textual data, are positioned in a shared embedding space. This is often achieved through contrastive learning methods [11, 12, 15, 26, 43]. Particularly in the video-text domain, approaches such as [2, 8, 41] aim to align these modalities in a more fine-grained manner.

## 3 METHOD

In this section, we provide an in-depth explanation of our approach that centers on the global text-anchor. As shown in Fig. 2, our framework integrates four components: feature extraction, cross-modal interaction, fine-grained alignment loss, and prediction. We first outline the task while detailing the feature extraction process in Sec. 3.1. We then introduce the core method of our framework in Sec. 3.2, where the concept of the global text anchor is introduced to encapsulate the holistic textual context. In Sec. 3.3, we detail

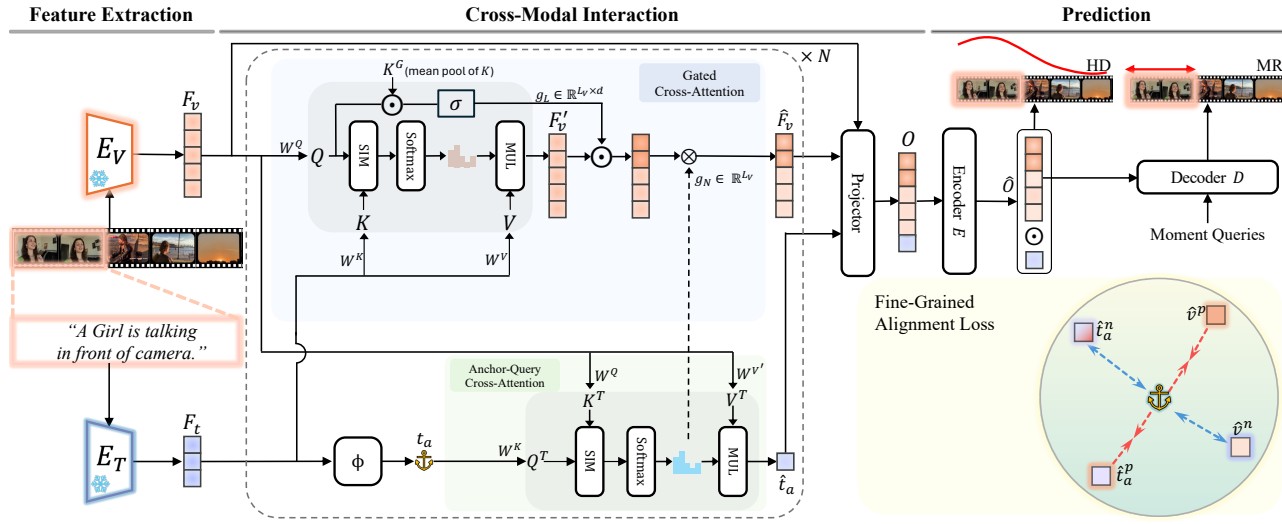

Figure 2: The pipeline of our framework consists of four components: feature extraction, cross-modal interaction, fine-grained alignment loss, and prediction. First, we extract visual and text features with frozen pre-trained encoders. Since the task requires cross-modal understanding and suppression of irrelevant information, we incorporate the gated cross-attention mechanism for the cross-modal interaction. The encoded features of cross-modal interaction are leveraged through the fine-grained alignment loss, which guides the model to enhance cross-modal alignment. Finally, the visual and textual representations from this aligned embedding space are fed into the prediction section to produce task-specific outputs.

two critical cross-attention mechanisms—gated cross-attention and anchor-query cross-attention. Both mechanisms utilize the global text anchor to enhance the interaction between the video content and the text query. We then present newly devised fine-grained alignment losses in Sec. 3.4, aimed at improving the alignment between video content and text queries. Finally, we detail the inference procedure, explaining how our model concurrently predicts for both Moment Retrieval (MR) and Highlight Detection (HD).

### 3.1 Preliminaries

Given a video $V \in \mathbb{R}^{L_v \times H \times W \times 3}$, consisting of $L_v$ frames sampled from the original video at specific intervals and a natural language text query $T$ comprising $L_t$ tokens, the objective is to identify all relevant moments $\{m_n = (m_{c_n}, m_{\sigma_n})\}_{n=1}^{N}$, where $m_{c_n}$ denotes the center coordinate of a moment and $m_{\sigma_n}$ represents its width. Additionally, the model predicts a frame-level saliency score $\{s_i\}_{i=1}^{L_v}$ concurrently. For feature extraction, following previous works [13, 17, 22, 36, 42], we employ pre-trained encoders $E_V$ and $E_T$ to extract visual features $F_v = [v_1, v_2, \ldots, v_{L_v}] \in \mathbb{R}^{L_v \times d_v}$ and text features $F_t = [t_1, t_2, \ldots, t_{L_t}] \in \mathbb{R}^{L_t \times d_t}$, respectively. Separate Multi-layer Perceptron (MLP) are used to project the video and text features into a shared embedding space of the same dimension $d$. In the following sections, we describe in detail our contributions of cross-modal interaction, fine-grained alignment loss, and how we train our model with our proposed methods.

### 3.2 Global Text Anchor

The primary goal of the VTG task is to identify video sequences that correspond to the *entire* query. In order to do this effectively,

an essential aspect of VTG is that it needs to capture the *overall meaning of a given text query*. Therefore, we propose to use a global text anchor that embodies a holistic understanding of the sentence, as emphasized in our title: *Let Me Finish My Sentence*. Adopting the global text query as an anchor offers two main advantages: (1) the text modality is generally less noisy compared to the visual modality due to its discreteness, and (2) we can effectively leverage prior knowledge from language models trained on a large corpora, which are well-generalized across various domains. We employ a global mean pooling operation, denoted as $\phi$, to derive a global text anchor from token-level text representations. This anchor is then integrated into our cross-modal interaction module and is further detailed in the following section.

### 3.3 Cross-Modal Interaction

As mentioned, we hypothesize the need for holistic understanding of the text query so that the model may take into account the entire text query instead of being biased to certain words in the query. Hence, we propose and introduce a global text anchor to address the aforementioned issues in these ways: 1) suppressing non-essential video frames through a gating mechanism with the global text anchor, 2) enhancing cross-modal alignment between video content and global textual information. The versatile use of the global text-anchor enhance the contextual relevance of video-text interactions and cross-model understanding.

**Gated Cross-Attention.** As shown in Fig. 2, we extend the conventional cross-attention mechanism [40] by incorporating *local* and *non-local* gates. The standard cross-attention is defined as:

$$\text{Attention}(Q, K, V) = \text{softmax}\left(\frac{QK^T}{\sqrt{d_k}}\right)V, \quad (1)$$

where $Q$, $K$, and $V$ represent the query, key, and value matrices respectively, and $d_k$ is the dimensionality of the key. In our model, this formula becomes:

$$\text{Attention}(Q, K, V) = \text{Attention}(W^Q F_v, W^K F_t, W^V F_t) = F_v', \quad (2)$$

where $W^Q$, $W^K$, and $W^V$ are learned weight that project the video features ($F_v$) and text features ($F_t$) into the query, key, and value.

The local gate weight, denoted by $g_L$, employs element-wise multiplication and sigmoid activation to assess the relevance between the video clip and the text query, utilizing the global key $K^G \in \mathbb{R}^d$. This key $K^G$ is derived by mean pooling $K$ across time dimensions:

$$g_L = \sigma \left( W_q^g Q \odot W_k^g K^G \right) \in \mathbb{R}^{L_v \times d}, \quad (3)$$

where $W_q^g$ and $W_k^g$ are trainable weights applied to the query $Q$ and the global key $K^G$, respectively, to capture channel-wise relevance between single frame and global text. The output, $g_L$, modulates the feature vector $V'$ through element-wise multiplication, resulting in a relevance-enhanced feature representation $V_L'$:

$$V_L' = g_L \odot F_v'. \quad (4)$$

The non-local gate weight $g_N$, designed for broad relevance assessment across video clips, modifies the interaction to enhance contextually pertinent video frames:

$$\hat{F}_v = g_N \odot V_L' = \{\hat{v}_1, \hat{v}_2, \ldots, \hat{v}_{L_v}\}. \quad (5)$$

This mechanism, $g_N \in \mathbb{R}^d$, computes min-max normalized attention scores between the global text anchor and video frames, effectively weighting the relative significance of each video frame in relation to the global text. It serves as an effective filter that emphasizes frames with substantial contextual relevance while suppressing the less relevant ones.

Through the integration of both local and non-local gates, our model emphasizes clips closely aligned with the text query while minimizing the influence of non-relevant clips in subsequent steps.

**Anchor-Query Cross-Attention.** Here, the global text anchor $t_a$, obtained through the mean pooling operator $\phi(F_t)$ is encoded into the visually enriched text-query. The video clip representations $F_v$ are adapted as both keys and values within the cross-attention mechanism. While the gated cross-attention approach learns the alignment between individual video frames and text tokens, the anchor-query cross-attention addresses the relative relevance between the global text representation and each of the video frames. The operational equation is defined as follows:

$$\text{Attention}(Q_a, K_a, V_a) = \text{Attention}(W^K t^a, W^Q F_v, W^{V'} F_v)$$
$$= \text{softmax}\left(\frac{Q_a K_a^T}{\sqrt{d_k}}\right) V_a = \hat{t}_a, \quad (6)$$

In this configuration, $W^Q$ and $W^K$ serve as projection layers for video and text modalities within the gated cross-attention mechanism, respectively. They aid in establishing the understanding between video content and textual information. Since $\text{softmax}\left(\frac{Q_t K_t^T}{\sqrt{d_k}}\right)$ represents the similarity score between the text query and video frames, we represent this calculation as $g_N$.

The two parts in the cross-modal interaction module refine the intermediate features based on cross-modal understanding. The module guides the model to emphasize the frames which are contextually relevant to the text query.

## 3.4 Fine-Grained alignment loss

Understanding of the fine-grained correlation between text queries and video clips is essential for MR and HD tasks. Moreover, the validity of the local and non-local gates depends on the reliability of cross-modal understanding. To address this, we propose two loss functions to better learn cross-modal alignment: 1) preserving the consistency of the global text representation before and after the anchor-query cross-attention step, and 2) inducing the model to capture the frame-level relevance between the visual frame and global text representation.

**Clip-Level Consistency Loss.** Suppose we are given a mini-batch of feature pairs $\{(F_v^i, F_t^i)\}_{i=1}^B$, where $B$ refers to the size of the mini-batch. The loss function incorporates the global representation $t_a^i = \phi(F_t^i)$ of text query $F_t^i$, and the output of the anchor-query cross-attention layer $\psi$, $\hat{t}_a^{ij} = \psi(t_a^i, F_v^j)$. We employ $\hat{t}_a^p$ and $\hat{t}_a^n$, positive and negative visual-enriched global text representation from paired ($i = j$) and unpaired ($i \neq j$) video-text respectively. The proposed loss function is designed to minimize the distance between the global text anchor $t_a$ and $\hat{t}_a^p$, while maximizing the distance between $t_a$ and $\hat{t}_a^n$, expressed as:

$$\mathcal{L}_{\text{clip}} = -\frac{1}{B} \sum_{i=1}^B \log\left(\frac{\exp(\hat{t}_a^{ii} \cdot t_a^i)}{\sum_{j=1}^B \exp(\hat{t}_a^{ij} \cdot t_a^i)}\right) \quad (7)$$
$$- \frac{1}{B} \sum_{j=1}^B \log\left(\frac{\exp(\hat{t}_a^{jj} \cdot t_a^j)}{\sum_{i=1}^B \exp(\hat{t}_a^{ij} \cdot t_a^j)}\right). \quad (8)$$

Optimizing the model with this objective enhances the cross-modal alignment between the text anchor $t_a$ and semantically relevant video clips. By minimizing the distance between $t_a$ and the visually enriched text query $\hat{t}_a^p$, which incorporates both relevant and irrelevant video frames, the approach guides the text anchor's attention strongly towards semantically relevant video frames.

The training objective aims to refine the video clip representation $\hat{F}_v$, derived from the gated cross-attention, to align closely with the text anchor $t_a$ for relevant text queries, and to diverge when the queries are irrelevant. This method conditions the model to enhance the semantic correlation between video clips and the corresponding text query, ensuring their representations in the embedding space accurately reflect their contextual relevance.

**Frame-Level Relevance Losses.** This loss function refines the representation of video clips $\hat{F}_v$, which has been processed through gated cross-attention, by optimizing the alignment between video frames and the text anchor. Specifically, it enhances the similarity between relevant video frames $\hat{v}^p$ and the text anchor $t_a$, while reducing the similarity with irrelevant frames $\hat{v}^n$. This loss guides the model to learn the fine-grained correlation between visual frames and the corresponding text query, ensuring their representations in the embedding space accurately reflect their contextual relevance.

The similarity score between the i-th frame $\hat{v}_i$ and the text anchor $t_a$ is given by $D^i = \sigma(\hat{v}_i \cdot a)$, where $\sigma$ denotes a sigmoid that incorporates the similarity score. The loss function then is:

$$\mathcal{L}_{\text{frame}} = \sum_{i=1}^{L_v} C^i \log(D^i) + (1 - C^i) \log(1 - D^i) \qquad (9)$$

Here, $C^i$ is a binary indicator reflecting whether the i-th clip is relevant (1) or irrelevant (0) to the text query.

## 3.5 Prediction and Losses

In our cross-modal interaction module, which consists of $N$ transformer layers, we aim to generate a composite representation. This is achieved by channel-wise concatenating the intermediate outputs from each layer $O_l$ with the input video features $F_v$, and then projecting this concatenation into a $d$-dimensional space using a linear projection layer $f$:

$$O' = f(\text{Concat}_{\text{channel}}(F_v, O_1, O_2, \ldots, O_N)) \in \mathbb{R}^{L_v \times d}. \qquad (10)$$

Subsequently, the output $\hat{t}_a$ is concatenated with $O$ in a temporal manner to form the basis for a query-dependent adaptive classifier [22, 35]:

$$O = \text{Concat}_{\text{temporal}}(O', \hat{t}_a) \in \mathbb{R}^{(L_v+1) \times d}. \qquad (11)$$

This concatenated output, denoted as $O$, is considered the final feature set of the cross-modal interaction module.

**Highlight Prediction.** The final feature set $O$ is processed through a transformer encoder $E$. Separate Multi-layer Perceptron (MLP) is used to project the video and text features into a shared embedding space of the same dimension $d$, producing $\hat{O} = \{\hat{o}_1, \ldots, \hat{o}_{L_v}, \hat{t}_a'\}$. Following QD-DETR [9, 22], we calculate a vector of saliency scores $S \in \mathbb{R}^{L_v}$ for every frame in the video as follows:

$$S_i = \frac{w_s \cdot \hat{t}_a' \cdot (w_v \cdot \hat{o}_i)}{d}, \quad \text{for } i = 1, \ldots, L_v, \qquad (12)$$

where $w_s$ and $w_v$ are learnable parameters applied to the query representation and each video frame representation respectively.

**Moment Retrieval Prediction.** Following decoder strategies from prior research [19, 22, 36], we utilize dynamic anchor boxes to represent moment queries (which is clearly separate from text queries). Together with the output $\hat{O} = \{\hat{o}_1, \ldots, \hat{o}_{L_v}\}$, this input is provided to the decoder $D$, resulting in moment features $Q$. $Q$ undergoes processing via a Multi-Layer Perceptron (MLP) and a sigmoid activation to generate predictions for moment dimensions ($\hat{m}$), yielding $M$ moment predictions. Concurrently, another linear layer equipped with a softmax activation categorizes each predicted moment as foreground or background ($\hat{p}$).

**Highlight Loss.** The highlight loss $L_{hl}$ is comprised of a margin contrastive loss $L_{\text{margin}}$ and a rank-aware contrastive loss $L_{\text{rank}}$. Margin loss contrasts high and low scoring frames within ($t_{\text{high}}$, $t_{\text{low}}$) and outside ($t_{\text{in}}$, $t_{\text{out}}$) ground-truth moments, given by:

$$\mathcal{L}_{\text{margin}} = \max(0, \Delta + S(t_{\text{low}}) - S(t_{\text{high}})) + \max(0, \Delta + S(t_{\text{out}}) - S(t_{\text{in}})), \qquad (13)$$

with $\Delta$ denoting the margin. Following QD-DETR [22], the rank-aware loss is:

$$\mathcal{L}_{\text{rank}} = -\sum_{r=1}^{R} \log \frac{\sum_{x \in X_r^{\text{pos}}} \exp(S_x/\tau)}{\sum_{x \in (X_r^{\text{pos}} \cup X_r^{\text{neg}})} \exp(S_x/\tau)}, \qquad (14)$$

where $X_r^{\text{pos}}$ and $X_r^{\text{neg}}$ are the indexes of positive and negative frames within the $r$-th rank group, respectively, and $\tau$ is a temperature scaling parameter. The total highlight loss is as follows:

$$\mathcal{L}_{\text{hd}} = \mathcal{L}_{\text{margin}} + \mathcal{L}_{\text{rank}}. \qquad (15)$$

**Moment Retrieval Loss.** To address the set prediction challenge in moment retrieval without a direct one-to-one correspondence between ground truth and predictions, we apply the Hungarian matching algorithm. This algorithm pairs ground truth moments with predictions, where $\hat{\sigma}(i)$ indexes the predicted moment matched to the $i$-th ground truth moment. The span loss for matched pairs is as follows:

$$\mathcal{L}_{\text{span}}(m_i, \hat{m}_{\hat{\sigma}(i)}) = \lambda_{\text{L1}} \|m_i - \hat{m}_{\hat{\sigma}(i)}\| + \lambda_{\text{iou}} \mathcal{L}_{\text{iou}}(m_i, \hat{m}_{\hat{\sigma}(i)}), \quad (16)$$

incorporating the generalized IOU loss [28]. The moment retrieval loss combines classification and span losses:

$$\mathcal{L}_{\text{mr}} = \sum_{i=1}^{N} [-\lambda_{\text{cls}} \log \hat{p}_{\hat{\sigma}(i)}(c_i) + I(c_i \neq \varnothing) \mathcal{L}_{\text{span}}(m_i, \hat{m}_{\hat{\sigma}(i)})], \quad (17)$$

where $I(\cdot)$ applies span loss only to non-empty ground truth moments.

**Overall Loss.** The overall loss is defined as:

$$\mathcal{L} = \mathcal{L}_{\text{hd}} + \mathcal{L}_{\text{mr}} + \lambda_{\text{clip}} \mathcal{L}_{\text{clip}} + \lambda_{\text{frame}} \mathcal{L}_{\text{frame}} \qquad (18)$$

where the coefficients $\lambda_{\text{clip}}$ and $\lambda_{\text{frame}}$ are parameters that balance the contribution of clip-level and frame-level losses to the overall loss, respectively.

## 4 EXPERIMENTS

In this section, we outline the experimental setup and list details on the dataset, evaluation metrics, and implementation specifics as discussed in Sec. 4.1. Then, we compare the performance of our framework with established baselines in Sec. 4.2 and show a detailed ablation study in Sec. 4.3. Lastly, we show qualitative results showcasing the effectiveness of our approach in Sec. 4.4.

### 4.1 Experimental Setup

**Datasets.** As our task is to detect highlights while retrieving moments, we use commonly used MR and HD dataset QVhighlights [13] as our main benchmark. The QVhighlights dataset currently stands as the sole dataset available to test both MR and HD tasks concurrently and comprises of over 10,000 YouTube videos accompanied by human-written, free-form text queries. For moment labels, each video-text pair is annotated with one or more relevant moments, and highlight labels are provided with 5-scale saliency scores (ranging from 1 being very bad to 5 being very good). Additionally, in order to ensure a fair benchmark for evaluation, the performance on the test set is assessed exclusively through submissions to the QVhighlights server.[1] In addition to this, to further test the efficacy of our method, we evaluate it on other VTG datasets, namely

---

[1] https://codalab.lisn.upsaclay.fr/competitions/6937

**Table 1: Experimental results on the QVHighlights test split, comparing performance in Moment Retrieval (MR) and Highlight Detection (HD). All models listed utilized uniform video (SlowFast and CLIP) and text features (CLIP).**

| Method | MR | | | | | HD | |
|---|---|---|---|---|---|---|---|
| | R1 | | mAP | | | ≥Very Good | |
| | @0.5 | @0.7 | @0.5 | @0.75 | Avg. | mAP | HIT@1 |
| XML+ [14] ECCV2020 | 46.69 | 33.46 | 47.89 | 34.67 | 34.90 | 35.38 | 55.06 |
| Moment-DETR [13] NeurIPS2021 | 52.89 | 33.02 | 54.82 | 29.40 | 30.73 | 35.69 | 55.60 |
| UMT [21] CVPR2022 | 56.23 | 41.18 | 53.83 | 37.01 | 36.12 | 38.18 | 59.99 |
| MomentDiff [16] NeurIPS2023 | 57.42 | 39.66 | 54.02 | 35.73 | 35.95 | - | - |
| MH-DETR [45] ACM MM2023 | 60.05 | 42.48 | 60.75 | 38.13 | 38.38 | 38.22 | 60.51 |
| QD-DETR [22] CVPR2023 | 62.40 | 44.98 | 62.52 | 39.88 | 39.86 | 38.94 | 62.40 |
| UniVTG [17] ICCV2023 | 58.86 | 40.86 | 57.60 | 35.59 | 35.47 | 38.20 | 60.96 |
| TR-DETR [36] AAAI2024 | 64.66 | 48.96 | 63.98 | 43.73 | 42.62 | 39.91 | 63.42 |
| UVCOM [42] CVPR2024 | 63.55 | 47.47 | 63.37 | 42.67 | 43.18 | 39.74 | 64.20 |
| **Ours** | **65.95** | **49.74** | **65.82** | **44.14** | **43.57** | **40.27** | **65.60** |

Charades-STA [6] and TACoS [27]. Charades-STA features 9,848 videos with 16,128 query-moment pairs focusing on indoor activities. TACoS comprises 127 videos annotated specifically for cooking scenarios.

**Evaluation Metrics.** We follow the conventions established in previous research [10, 13, 17, 21, 22, 36, 42, 45]. In MR, we apply Recall@1 (R@1) at IoU thresholds of 0.5 and 0.7. We also calculate mean Average Precision (mAP) for IoU thresholds [0.5 : 0.05 : 0.95]. For HD, we measure mAP and HIT@1, where HIT@1 is determined by the hit ratio of the clip with the highest score.

**Baseline Architecture.** Our framework is built upon the QD-DETR [22] architecture, which is a widely used baseline in Video Temporal Grounding due to its effective use of cross-attention layers for injecting text query information into video frames. We build upon this baseline while retaining its standard cross-attention mechanism, decoder structure, and rank-aware loss. Our method introduces novel approaches aimed to effectively leverage the holistic context of text queries for improved video frame selection and alignment.

**Implementation Details.** We configure the number of layers in the transformer encoder $E$ and decoder $D$ as 3. We set the cross-modal interaction layer count to 2. The loss balancing parameters are established as $\lambda_{L1} = 10$, $\lambda_{iou} = 1$, $\lambda_{cls} = 4$, $\lambda_{frame} = 1$, and $\lambda_{clip}$ is adjusted to 1 for QVHighlights and Charades-STA, and 0.6 for TACoS. We set the batch size to 32 and the learning rate (LR) to 0.0001 for QVHighlights, maintain a batch size of 32 with an LR of 0.0002 for Charades. We adjust the batch size of 16 with an LR of 0.0002 for TACoS following previous works. Across all datasets, training proceeds for 200 epochs with a learning rate reduction at epoch 100, using the Adam optimizer. Additionally, for all datasets, we set the hidden dimension $d$ to 256 and the number of moment queries $M$ to 10. Whereas otherwise stated, we employ a pre-trained SlowFast and CLIP [26] model for video feature extraction. Specifically for Charades-STA, additionally features were extracted using VGG [32], C3D [39], and GloVe [25]. All models were trained on a single NVIDIA RTX 4090 with an average training time of 3 hours for all 200 epochs on our machines.

## 4.2 Main Result

We compare how our method performs in relation to recent state-of-the-art methods for MR and HD and summarize our findings in the following sections.

**Results on QVHighlights.** In Table 1, we list the experimental results of our method as well as other established methods on the QVHighlights dataset. Our method diverges from XML [14], which adopts a proposal-free strategy, and aligns with the transformer-based and end-to-end trainable nature of current baselines [10, 13, 16, 17, 21, 22, 36, 42, 45]. MH-DETR [45] and QD-DETR [22] focus more on cross-modal interaction before transformer encoder and TR-DETR [36] explores the inherent reciprocity between MR and HD, and UVCOM [42] is tailored to address the unique demands of both MR and HD tasks effectively. Although EaTR [10] is a fairly recent work, we do not list them in our main table as they do not evaluate on the QVHighlights test split. Our model capitalizes on global text semantics and novel loss functions within a refined cross-modal interaction framework, surpasses all compared methods. Notably, our method outperforms the baseline model, QD-DETR [22], in MR by achieving a 3.5% improvement in R@1 at IoU 0.5, 4.76% improvement at IoU 0.7, and 3.7% increase in average mAP, and in HD by 1.63 mAP and 3.20 in HIT@1. Our method outperforms the most recent method, UVCOM [42], in MR by 2.40% R@0.5, 2.24% R@0.5, and 0.39% average mAP, and in HD by 0.50 mAP and 1.40 HIT@1 respectively. We find that our method outperforms all previous baselines across all metrics in both MR and HD, to the best of our knowledge, setting the new state-of-the-art.

**Results on Charades-STA and TACoS.** To test the generalizability and efficacy of our method, we extend our experimentation to other VTG benchmarks such as Charades-STA [6] and TACoS [27]. On top of our comparisons with transformer-based models, we further test our approach against prior proposal-based approaches [49, 51]. As evidenced in Tables 2 and 3, our method surpasses previous techniques, establishing new state-of-the-arts on these benchmarks. Particularly on Charades-STA, our approach demonstrates its robustness by consistently outperforming existing methods across a variety of backbones, including learned multi-modal features from CLIP [26], as well 2D and 3D features from

**Table 2: Experimental results on the Charades-STA test split. All models employed uniform features for fairness in evaluation, with 'SF+C, C' denoting SlowFast and CLIP for video and CLIP for text, respectively, 'VGG, Glove' indicating VGG features for video and GloVe embeddings for text, and 'C3D, Glove' representing C3D features for video and GloVe embeddings for text.**

| Method | feat | R@0.5 | R@0.7 |
|---|---|---|---|
| Moment-DETR [13] NeurIPS2021 | SF+C, C | 53.63 | 31.37 |
| MomentDiff [16] NeurIPS2023 | SF+C, C | 55.57 | 32.42 |
| QD-DETR [22] CVPR2023 | SF+C, C | 57.31 | 32.55 |
| UniVTG [17] ICCV2023 | SF+C, C | 58.01 | 35.65 |
| TR-DETR [36] AAAI2024 | SF+C, C | 57.61 | 33.52 |
| UVCOM [42] CVPR 2024 | SF+C, C | 59.25 | 36.64 |
| **Ours** | SF+C, C | **60.73** | **39.49** |
| MAN [49] CVPR2019 | VGG, GloVe | 41.24 | 20.54 |
| 2D-TAN [51] AAAI2020 | VGG, GloVe | 40.94 | 22.85 |
| MomentDiff [16] NeurIPS2022 | VGG, GloVe | 51.94 | 28.25 |
| QD-DETR [22] CVPR2023 | VGG, GloVe | 52.77 | 31.13 |
| TR-DETR [36] AAAI2024 | VGG, GloVe | 53.47 | 30.81 |
| UVCOM [42] CVPR2024 | VGG, GloVe | 54.57 | 34.13 |
| **Ours** | VGG, GloVe | **56.56** | **37.28** |
| IVG-DCL [24] CVPR2021 | C3D, GloVe | 50.24 | 32.88 |
| MomentDiff [16] NeurIPS2023 | C3D, GloVe | 53.79 | 30.18 |
| QD-DETR [22] CVPR2023 | C3D, GloVe | 50.67 | 31.02 |
| **Ours** | C3D, GloVe | **54.78** | **35.13** |

**Table 3: Experimental results on the TACoS test split. All models utilized uniform video (SlowFast and CLIP) and text features (CLIP).**

| Method | R@0.3 | R@0.5 | R@0.7 | mIoU |
|---|---|---|---|---|
| 2D-TAN [51] AAAI2020 | 40.01 | 27.99 | 12.92 | 27.22 |
| VSLNet [50] ACL2022 | 35.54 | 23.54 | 13.15 | 24.99 |
| Moment-DETR [13] NeurIPS2021 | 37.97 | 24.67 | 11.97 | 25.49 |
| UniVTG [17] ICCV2023 | 51.44 | 34.97 | 17.35 | 33.60 |
| UVCOM [42] CVPR2024 | - | 36.39 | 23.32 | - |
| **Ours** | **52.04** | **39.12** | **23.62** | **36.09** |

VGG [32] and C3D [39], respectively. Notably, with features combining SlowFast [5] and CLIP (SF+C, C), our method surpasses the recent state-of-the-art model, UVCOM [42], by achieving improvements of 1.48% in R@1 at an IoU of 0.5 and 2.85% at an IoU of 0.7. On TACoS, our method once again outperforms all previous baselines, affirming its effectiveness across diverse domains and datasets.

## 4.3 Ablation Studies

To understand the individual components of our framework and its effects, we present a series of ablation studies on QVHighlights validation split.

**Gated Cross-Attention.** We analyse the significance of employing both local and non-local gates within our gated cross-attention framework and present it in Table 4. Implementing the non-local gate alone enhances performance in MR and HD tasks, underscoring

**Table 4: Ablation study results on QVHighlights val split regarding gated cross-attention mechanisms. The 'local' and 'non-local' columns refer to the use of local and non-local gates, respectively.**

| local | Non-local | MR R1 @0.5 | MR R1 @0.7 | mAP Avg. | HD ≥Very Good mAP |
|---|---|---|---|---|---|
| | | 63.29 | 48.45 | 42.22 | 39.69 |
| ✓ | | 62.45 | 48.90 | 42.5 | 39.82 |
| | ✓ | 65.87 | 49.29 | 43.61 | 40.79 |
| ✓ | ✓ | **67.61** | **50.65** | **44.8** | **40.98** |

**Table 5: Highlight Detection performance on QVHighlights val split, showcasing the validity of non-local gate weights in gated cross-attention. † denotes intermediate outputs assessed directly for HD.**

| Method | HD ≥Very Good mAP | HD ≥Very Good HIT@1 |
|---|---|---|
| $g_N$ w/o Non-local[†] | 25.31 | 41.16 |
| $g_N^†$ | 35.83 | 56.71 |
| Moment-DETR [13] | 35.69 | 55.60 |
| **Ours** | **40.98** | **65.35** |

its utility. The synergy of local and non-local gates, however, yields the best outcome, underlining their collective importance in refining video-text alignment and enhancing prediction accuracy.

To further assess the validity of our non-local gate weight ($g_N$) within the gated cross-attention framework, we test to see if it can be used directly as a saliency score for Highlight Detection on the QVHighlights validation split. Table 5 demonstrates that using $g_N$ directly to predict saliency not only exceeds Moment-DETR's HD performance but also shows the utility of our gate mechanism in emphasizing relevant video sections in cross-modal interaction. Additionally, '$g_N$ w/o Non-local,' represents a scenario where $g_N$ is computed without applying the non-local gate, which corresponds to the the second row of Table 4, and this results in a significant drop in performance. This underlines the non-local gate's critical role in emphasizing relevant video frame and enhancing the accuracy of the similarity measure between global text and video frames.

**Fine-Grained Alignment Losses.** We further analyse the impact of the frame-level and clip-level similarity losses in Table 6. Implementing each loss individually offers noticeable improvements; however, integrating both simultaneously provides a substantial performance boost. Specifically, without these similarity losses, the model achieves lower scores across all metrics. With both losses applied, we observe a 5.29% increase in R@1 at IoU 0.5, a 3.3% rise at IoU 0.7, and a 2.88% improvement in mAP, highlighting their effectiveness in refining the model's capability to align video content with textual queries accurately.

**Global Text Anchor.** We also explore various methods for generating the global text anchor, which is pivotal to our framework as

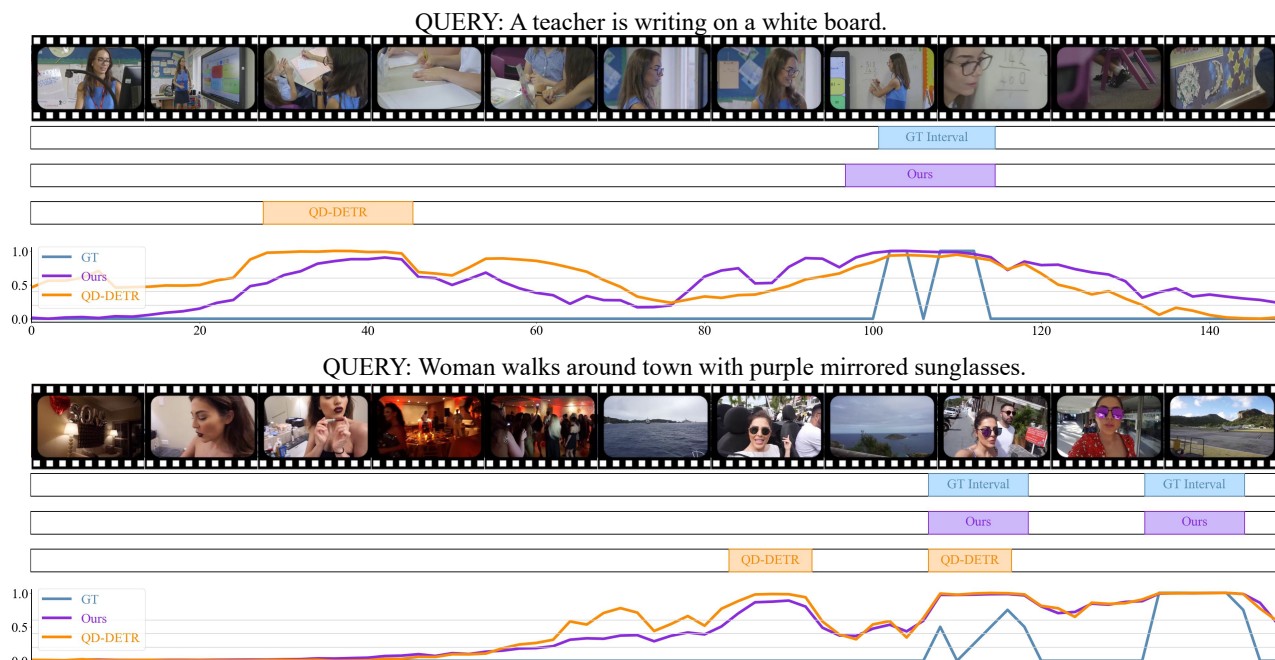

**Figure 3: Qualitative results of predictions on QVHighlights validation split. We show the effectiveness of our method compared to the baseline, QD-DETR. From top to bottom are the text queries, along with the predicted moments and highlights corresponding to each method.**

**Table 6: Ablation study results on QVHighlights val split on text-anchor similarity loss. The 'frame' and 'clip' columns denote the frame-level loss and clip-level loss, respectively.**

| frame | clip | MR R1 @0.5 | @0.7 | mAP Avg. | HD ≥Very Good mAP |
|-------|------|------|------|------|------|
| | | 62.32 | 47.35 | 41.92 | 39.02 |
| | ✓ | 65.48 | 49.23 | 43.20 | 40.46 |
| ✓ | | 63.48 | 48.13 | 43.43 | 39.56 |
| ✓ | ✓ | **67.61** | **50.65** | **44.80** | **40.98** |

**Table 7: Ablation study on various global text anchor generation methods on QVHighlights val split.**

| Method | MR R1 @0.5 | @0.7 | mAP Avg. | HD ≥Very Good mAP |
|--------|------|------|------|------|
| Max pooling | 64.45 | 49.55 | 44.07 | 40.48 |
| Weighted pooling | 65.87 | 49.29 | 43.61 | 40.79 |
| Transformer | 65.68 | 50.06 | 44.24 | 40.30 |
| Mean pooling | **67.61** | **50.65** | **44.80** | **40.98** |

it focuses attention on the relevant parts of the video corresponding to the text query. To determine the most effective approach, as detailed in Table 7, we compared mean pooling, max pooling, weighted pooling, and the use of a transformer layer. We find that mean pooling outperforms other methods. This underscores the effectiveness of a simple yet powerful mean pooling strategy in capturing the holistic semantics of the text query for VTG.

### 4.4 Qualitative Results

We show qualitative results in Fig. 3 compared with the baseline model, QD-DETR. By adopting a holistic approach to understanding text queries, our method consistently identifies moments that fully align with the intent of the textual queries. This approach allows our model to capture the essence of the entire query, preventing the oversight of integral query components such as 'a white board'

and 'walks around', which QD-DETR sometimes overlooks. This underscores the advantage of our method's integration of global text semantics for more accurate video grounding.

### 5 CONCLUSION

In this work, we present a novel approach to Video Temporal Grounding (VTG) that emphasizes holistic understanding of text queries and suppresses irrelevant visual frames. By integrating the entire query text into a global representation and employing visual frame-level gate mechanisms within a cross-modal interaction framework, our approach significantly enhances the alignment of text queries with accurate video segments. We demonstrate state-of-the-art performance on several VTG benchmarks, highlighting the importance of considering the entire text query and selectively focusing on relevant video.

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
