# OpenReview forum: "Let Me Finish My Sentence: Video Temporal Grounding with Holistic Text Understanding"
_acmmm.org/ACMMM/2024/Conference — MM2024 Poster_

### Official Review · Reviewer_5YyS · 2024-05-16

**Rating:** 4
**Confidence:** 3

**Summary:**

This paper introduces a novel framework for Video Temporal Grounding that emphasizes holistic text understanding and selective attention to relevant video frames. The proposed method integrates global text semantics and frame-level gate mechanisms to enhance alignment between text queries and video segments, leading to improved accuracy in video grounding tasks. Qualitative results demonstrate the effectiveness of the approach in capturing text query essence and accurately identifying corresponding video moments.

The framework includes feature extraction, cross-modal interaction, fine-grained alignment loss, and prediction components, incorporating global text semantics and gated cross-attention mechanisms for enhanced video-text interaction. Fine-grained alignment losses are introduced to refine the alignment between video content and text queries, improving semantic correlation between the two modalities. The method outperforms existing models in Video Temporal Grounding benchmarks.

**Strengths:**

1.The framework introduces a novel method that emphasizes holistic text understanding and selective attention to relevant video frames, showcasing originality in addressing video temporal grounding tasks.

2.By incorporating global text semantics and frame-level gate mechanisms, the model improves alignment between text queries and video segments, leading to more accurate video grounding results.

3.The method outperforms state-of-the-art models on multiple benchmarks, demonstrating its effectiveness in capturing the essence of text queries and accurately identifying corresponding video moments.

4.Leveraging text queries for improved video frame selection and alignment, the framework showcases promising results in moment retrieval and highlight detection tasks, indicating the practical benefits of considering text queries in video temporal grounding.

**Limitations:**

1.The introduction of global text-anchor and fine-grained alignment loss functions may increase the complexity of the model, potentially leading to longer training times and higher computational costs.

2.While the method outperforms existing models on multiple benchmarks, its generalizability to diverse datasets and real-world applications may require further validation and testing.

**Suitability:**

3

---

### Official Review · Reviewer_6bx1 · 2024-05-21

**Rating:** 3
**Confidence:** 3

**Summary:**

This paper proposes a new pipeline to deal with the video temporal grounding task. The authors consider that existing works usually neglect the holistic understanding of the query sentence and miss out on the global meaning. As a result, the paper introduces two strategies, frame-level gate mechanism and cross-model alignment loss to solve these problems. The results on three popular banchmarks show the effectiveness of the proposed strategies.

**Strengths:**

1. The paper is well-written and easy to read. The author(s) introduce a practical pipeline to solve the video grounding task.
2. The results on QVHighlights, Charades-STA, and TACoS surpass the state-of-the-art models and show the effectiveness of the proposed method.
3. The author gives detailed ablation studies to verify the effect of each component of the framework.

**Limitations:**

1. The proposed local and non-local attention gates somehow lack novelty. Some existing works also consider local and global text features like LGI, TRM[1]. The clip-level consistency loss is also similar to the loss proposed by PLPNet[2] and LGI. Comparisons with these works should be contained in the paper and many references are missing.
2. Could this method also work on ActivityNet Caption dataset? More experiments and comparisons should be conducted.
3. As shown in Fig.3, the visualized results show the attention score of each frame. However, for some of those frames that are not contained in the range of ground truth, the predicted scores are still high or nearly equal to the baseline model. It cannot reflect the effects of the proposed modules.

[1] Phrase-level Temporal Relationship Mining for Temporal Sentence Localization. 2023.
[2] Phrase-level Prediction for Video Temporal Localization. 2022.

**Suitability:**

3

---

### Official Review · Reviewer_QBnb · 2024-05-25

**Rating:** 3
**Confidence:** 3

**Summary:**

The article introduces the goal of Video Temporal Grounding (VTG), showing the critical role of holistic text understanding in Video Temporal Grounding. To address this problem, the authors proposes two key contributions: introducing a visual frame-level gating mechanism and cross-modal alignment loss. These solutions help improve the model's performance in VTG tasks and emphasize the importance of holistic text understanding in guiding the model to focus on the semantically important parts within the video.

**Strengths:**

1. The article introduces visual frame-level gating mechanisms and cross-modal alignment loss, providing new insights for addressing issues in Video Temporal Grounding (VTG) tasks.
2. The article emphasizes the importance of holistic text understanding in guiding the model to focus on the semantically important parts within the video, which helps improve model performance.
3. The introduction of visual frame-level gating mechanisms and cross-modal alignment loss provides new insights for addressing issues in Video Temporal Grounding (VTG) tasks.
4. Extensive experiments are conducted, showing the effectiveness of the proposed method.

**Limitations:**

1. The paper introduces a cross-attention contribution that closely resembles existing work in the field, such as the cross-gated attended recurrent network described in previous studies [1,2], which also match natural sentences to video sequences for fine-grained interaction. Given these similarities, the claimed novelty of this idea is questionable.
2. While the cross-modal alignment loss is logically presented within the paper, its generality and applicability across different VTG models remain uncertain. The method appears to be highly sensitive to an array of hyper-parameters,  including different loss balancing parameters: $\lambda_{L1} = 10$, $\lambda_{iou} = 1$, $\lambda_{cls} = 4$, $\lambda_{frame} = 1$. Furthermore, the $\lambda_{clip}$ seems need to tuned in the TACoS dataset. More thorough discussion regarding the hyper-parameter tuning process is necessary to strengthen the paper's methodology.
3. The symbol $V_{L}^{'}$ in Formula 4 is not represented in the figure. The author should add appropriate labels, which can make the figure easier to understand.
4. The design of Figure 2 is overly complex, with numerous intersecting lines and unclear meanings, making the figure difficult to interpret.  Additionally, there is excessive white space that could be used more effectively to create a tighter, clearer presentation.  Simplifying or optimizing the diagram can significantly improve the clarity and effectiveness of the visual representation.

[1] Localizing Natural Language in Videos

[2] Fine-grained Iterative Attention Network for Temporal Language Localization in Videos

**Suitability:**

3

---

### Meta-Review · Area_Chair_eecN · 2024-07-02

**Recommendation:** Accept (Poster)
**Confidence:** 5

**Metareview:**

This paper receives two borderline rejects, and one borderline accept initially. The reviewers raise some questions regarding novelty, hyper-parameter tuning, presentation, results on ActivityNet Caption dataset, the visualized results, computational cost, generalizability, etc. Most of these questions are addressed in the rebuttal and recognized by reviewers. Eventually with the merit of the work, this paper receives one borderline reject, one borderline accept, one borderline accept (initial score without final update). Considering the author response to the reviewers' questions, the AC recommended to accept. Authors are encouraged to revise the paper according to the reviews.